# Prediction of Resistance Spot Welding Quality Based on BPNN Optimized by Improved Sparrow Search Algorithm

**DOI:** 10.3390/ma15207323

**Published:** 2022-10-20

**Authors:** Jianming Hu, Jing Bi, Hanwei Liu, Yang Li, Sansan Ao, Zhen Luo

**Affiliations:** 1School of Materials Science and Engineering, Tianjin University, Tianjin 300354, China; 2Tianjin Long March Launch Vehicle Manufacturing Co., Ltd., Tianjin 300462, China

**Keywords:** resistance spot welding, quality prediction, sparrow search algorithm, sine chaotic mapping, backpropagation neural network

## Abstract

Accurately predicting resistance spot welding (RSW) quality is essential for the manufacturing process. In this study, the RSW process signals of 2219/5A06 aluminum alloy under two assembly conditions (including gap and spacing) were analyzed, and then artificial intelligence modeling was carried out. To improve the performance and efficiency of RSW quality evaluation, this study proposed a multi-signal fusion method that was performed by combining principal component analysis and a correlation analysis. A backpropagation neural network (BPNN) model was optimized using the sine-chaotic-map-improved sparrow search algorithm (SSA), and the input and output of the model were the variables after multi-signal fusion and the button diameter, respectively. Compared with the standard BPNN model, the Sine-SSA-BP model reduced the MAE by 42.33%, MSE by 51.84%, and RMSE by 31.45%. Its R^2^ coefficient reached 0.6482, which is much higher than that of BP (0.2464). According to various indicators (MAE, MSE, RMSE, and R^2^), the evaluation performance of the Sine-SSA-BP model was better than that of the standard BPNN model. Compared with other models (BP, GA-BP, PSO-BP, SSA-BP, and Sine-PSO-BP), the evaluation performance of the Sine-SSA-BP model was best, which can successfully predict abnormal spot welds.

## 1. Introduction

Due to its high degree of automation, resistance spot welding (RSW) is widely used in connecting the automotive body-in-white and aerospace structures [1]. Applying electrode pressure on the upper and lower workpiece surfaces, and passing a current through the electrodes on both sides, can generate joule heat through the contact resistance of the workpieces’ bonding surface to melt and connect the workpieces [2]. The nugget is generated between the joint surfaces of the workpiece and cannot be directly observed, and the quality of RSW suffers from many disturbances in the manufacturing process, so it is crucial to monitor the quality of the buttons [3].

Destructive methods such as stripping and hammering are often used in factories to inspect the quality of buttons, which are too inefficient. Because the RSW process is a combined action of force, heat, electricity, magnetism, and flow, there is an inherent relationship between nugget formation and welding process signals [4]. Therefore, the quality of the joint can be predicted by various welding process signals.

Because the artificial neural network (ANN) is particularly suitable for solving nonlinear problems, it is often used in engineering to assist production [5]. Many researchers have used the ANN to predict weld quality. Chang et al. [6] predicted the leg length of asymmetric fillet root welds through a backpropagation (BP) neural network, and the error rate was less than 7%. Zhao et al. [7] used features extracted from power signals to establish a neural network (NN) model and successfully used it to evaluate the quality of spot welds. Wan et al. [8] successfully predicted the weld strength and nugget size of spot welding by using the features of the voltage signal as the input of the NN model. The authors also confirmed that the combination of dynamic resistance measurement and the NN model could effectively realize the quality monitoring of small-scale resistance spot welding [9]. Chen et al. [10] proposed a parallel strategy method based on the BP neural network to predict the RSW joint quality of automotive bodies; the R^2^ overall fit of the algorithm could reach 0.81–0.96. Panza et al. [11] constructed a NN model based on the characteristics of electrode displacement signals to predict the electrode contact area and further obtain the electrode degradation data. Pouraliakbar et al. [12] developed an BPNN model to predict the toughness of high-strength low-alloy steels, and the predicted values were in very good agreement with the measured ones, indicating that the model had a great ability for predicting the toughness of pipeline steels.

The BP neural network is widely used and favored by many scholars. However, it also has some defects. For example, its hyperparameters are difficult to determine, and the initial weights are too random. According to these characteristics, swarm intelligence algorithms can be used to optimize the neural network. In order to monitor the welding quality, Pashazadeh et al. [13] specified the optimum values of the welding time, the welding pressure, and the welding current by combining the artificial neural networks and the multi-objective genetic algorithm (GA). They used a genetic algorithm based on the fitness function of the ANN model to obtain the maximum joint strength. Hamidinejad et al. [14] used a genetic algorithm based on the fitness function of an ANN model to determine a set of process parameters. As a result, the maximum values of the strength of the RSW joints were obtained. Shojaeefard et al. [15] developed an ANN model to analyze the correlation between the friction stir welding parameters and the hardness of the welding joint, and they obtained the Pareto-optimal set by the particle swarm optimization (PSO) algorithm. Katharasan et al. [16] used the ANN to predict the weld bead geometry in the process of flux-cored arc welding, and they used the PSO algorithm to optimize the process parameters.

Moreover, the integration between artificial intelligence modeling and the meta heuristic optimizer has shown good application prospects in different engineering problems. Khoshaim et al. [17] built a multilayer perceptron model optimized by the gray wolf optimizer to predict the mechanical properties of friction-stir-welded aluminum alloys. Abushanab et al. [18] used the hunger games search optimizer to optimize a random vector functional link model, and the model was used to correlate the joint characteristics with the welding variables. Moustafa et al. [19] built an ANN model to predict the thermal efficiency and water yield of the solar still. The model was optimized by a meta-heuristic optimizer called humpback whale optimizer. Khoshaim et al. [20] optimized an ANN model by the flower pollination algorithm to model the residual stresses generated during dry turning of DT4E pure iron. Elsheikh et al. [21,22,23] developed three random vector functional link network models to predict the glass-fiber-reinforced epoxy composites drilling process, the responses of AISI 4340 alloy cutting process, and the ultrasonic welding of a polymeric material blend. The models were optimized by the parasitism-predation algorithm, a political optimizer, and a gradient-based optimizer, respectively. They also developed an ANN model optimized by a pigeon optimization algorithm to predict the induced residual stresses during turning of Inconel 718 alloy [24].

The sparrow search algorithm (SSA) is an emerging swarm intelligence and meta-heuristic algorithm proposed by Xue and Shen in 2020 and has been used by many researchers to solve intelligent manufacturing problems [25]. Many of them in nonwelding fields have compared the SSA with the GA and PSO [26,27]. The results showed that SSA has excellent optimization performance, such as a low error rate and good ability to resist over fitting. However, SSA also has problems such as declining population diversity and the tendency to fall into a local optimum. Considering the sine chaotic maps’ distribution uniformity and efficient convergence, this study uses an improved sparrow search algorithm (Sine SSA) to predict button diameter and realize an accurate online prediction of RSW joint quality. First, this paper analyzes the correlation between various welding process signals and button diameter, and the fusion of welding process signals is carried out. Sine mapping is then introduced to diversify the initial distribution of sparrows. Finally, the Sine-SSA-BP RSW joint quality evaluation model is built, and its reliability and stability are verified by the 10-fold cross-validation method. The main contributions and innovations of this paper are as follows:(1)This paper proposes a method of multi-signal fusion to obtain the integrated signal. First, the correlation analysis is carried out on the data of multiple RSW process signals after principal component analysis (PCA) dimension reduction. The data related to the button diameter are extracted and combined to realize multi-signal fusion. Further, the integrated signal is used to build the RSW prediction model, which contributes a lot to efficient quality monitoring.(2)For the first time, the emerging intelligent algorithm SSA is adopted to optimize a BPNN for the prediction of the RSW quality. Based on model output, Sine-SSA is used to optimize the weights and thresholds of the BPNN and to construct a Sine-SSA-BP prediction model to predict the button diameter.

## 2. Experimental

The experimental materials, equipment, and parameter settings are consistent with the authors’ previous study [28]. The materials used are 2219 and 5A06 aluminum alloy sheets, and their dimensions are 400 mm × 300 mm × 7 mm and 300 mm × 50 mm × 2 mm, respectively. Their chemical compositions and mechanical properties are given in Table 1 and Table 2, respectively.

During the manufacturing process, some special assembly conditions will be encountered, such as the welded structure with an arc surface or workpiece warping due to the influence of thermal stress. At this time, there will be a gap between the workpieces. In order to ensure the connection strength, the weld spots may be closely arranged. As shown in Figure 1, in order to study the joint quality of different fit-up welding conditions, the gap and spacing are set to simulate the practical welding conditions. The specific realization of the gap and spacing is shown in Figure 1. Silicone strips of different thicknesses are used to create different gaps between the two workpieces. In the experiment with different gaps, the spacing is fixed at 50 mm, as shown Figure 1a. In experiments with different spacing, the gap is fixed at 0 mm, as shown in Figure 1b. On the 2219 aluminum alloy sheet, there are 6 5A06 aluminum alloy sheets (as shown in the 6 dashed boxes in Figure 1b) symmetrically distributed along the length of the former. In addition, the starting welding point and the ending welding point have a distance of 50 mm from the edge of the workpiece. As shown in Table 3, there are 270 spot welds used in this study.

Figure 2 shows the experimental process parameters. The current setting mainly includes a welding current application phase and a tempering current application phase. The welding current is the main factor affecting the diameter of the nugget, and the nugget undergoes nucleation and growth phases during the period it is applied. The primary function of the tempering current is to ensure that the nugget will not produce solidification cracks and form a uniform structure. While applying welding current, the electrode force gradually increases and reaches a stable value. This setting benefits in obtaining a more considerable initial contact resistance and enough heat to melt the workpiece. Then, the electrode force enters the maintenance stage, and the extensive maintenance force can ensure that defects such as splashes and pores will not generate.

## 3. Results and Discussion

### 3.1. Signal Analysis

Figure 3 shows the partial signal curves collected during the experiment and they are preprocessed by filtering. The filtered curves are smoother than non-filtered curves, which can show the change in the process signal more intuitively.

It can be seen from Figure 3a that the welding current curves under different welding assembly conditions are basically the same, and the degree of discrimination visible to the naked eye is minimal. The reason is that the control system of the welding machine is very stable, and the welding current collected twice will not fluctuate violently with the change in the electrical characteristics of the workpiece. In the initial stage, the voltage increases as the current increases, as shown in Figure 3b. It is then abruptly decreased due to the softening of the workpiece resulting in a sudden decrease in the initial contact resistance. Next, the current rises again, and the voltage increases with it. A transient increase or decrease in current can cause a significant induced electromotive force in the voltage measurement circuit, as evidenced by the positive and negative voltage spikes in Figure 3b.

As shown in Figure 3c, the resistance curve has two peaks, the first due to the instantaneous drop in the initial contact resistance and the second due to applying the tempering current. Figure 3d shows that the root-mean-square (RMS) power curve is roughly the same as the current curve, except that the tempering stage produces a slight fluctuation due to the voltage fluctuation.

Figure 3e shows that the displacement signal drops steeply in the initial stage. Because the contact plane between the workpieces is relatively rough at this time, the initial contact resistance is considerable. Under the combined action of current and initial contact resistance between the workpieces, a large amount of joule heat is generated. The heat will cause the temperature to rise rapidly, which leads to the softening of the workpieces. Then, under the combined action of current and resistance, the temperature of the workpiece continues to rise, the workpiece is thermally expanded, and the electrode displacement begins to increase. After the tempering current is applied, the heat dissipation of the workpiece begins to play a leading role in the temperature change, and the displacement increase rate gradually slows down. The electrode force curve has a dip between 0.2 and 0.4 s, as shown in Figure 3f. This is due to softening of the workpiece, and then it reaches a maximum value during the tempering phase.

The above analysis can reflect that the changes in each welding process signal correspond to the growth process of the nugget, which contains information related to the quality of the welding button. Therefore, it is necessary to analyze all signals when predicting button quality.

### 3.2. Multi-Signal Fusion

Because six kinds of process signals are collected in the RSW process, it will undoubtedly have a large workload to analyze the characteristics of each kind of process signal. If the preset process parameters are changed, each signal’s feature extraction and analysis must be performed again. These two reasons are not conducive to efficient welding quality prediction, so this study uses PCA instead of feature extraction to achieve the purpose of data dimensionality reduction. PCA can reduce variables in a data set to a specified dimension. Moreover, it can reduce the computational workload and exclude the influence of the correlation between the feature data [29]. It does not require expert intervention in the prediction process, which can achieve the purpose of fast data processing. Moreover, after PCA, the components are sorted according to the size of the cumulative variance contribution rate, and the internal components of each signal after dimensionality reduction are linearly independent. This facilitates fast indexing of valuable components to predict weld quality efficiently.

Figure 4 shows the relationship between the cumulative variance contribution rate and the number of principal components after each signal’s PCA. In order to preserve the primary information of each signal as much as possible, the criterion for selecting the number of principal components is to make the cumulative variance contribution rate of each signal reach 99%.

When the correlation coefficient is less than 0.2, the correlation between the data is weak or nonexistent. Therefore, the correlation is analyzed between all the components after PCA and the button diameter y, and the components whose correlation coefficient is greater than 0.2 are taken out. The result is that the number of signal components related to the RSW joint’s diameter is the first three components of the current, the first four components of the voltage, the first two components of the power, and the first two components of the displacement (i.e., C1, C2, C3, V1, V2, V3, V4, P1, P2, D1, and D2, respectively). The resistance signal and the electrode force signal have no correlation with the button diameter.

The correlation between each signal component and button diameter in Figure 5 does not reach 0.5. That is, the correlation between all process signals and button diameter does not reach a strong correlation. This is because all the welding process signals in this study are collected from the same set of process parameters, and only the changes in welding conditions cause the fluctuations in the welding process signals. If welding conditions have little effect on the electrical properties of the workpiece, then the button diameter will also change little. Therefore, using only one of the signals to predict the joint quality does not provide enough information to build a predictive model. However, using all the components that correlate with button diameter as the input of the neural network can make full use of the information of each welding signal to achieve the purpose of multi-signal fusion.

A summary of the multi-signal fusion process is shown in Figure 6. First, the preprocessed single signals are dimensionally reduced by PCA. Then, correlation analysis is carried out between the dimensionality-reduced data and the output of the prediction model. Finally, the correlated data are integrated to obtain integrated signals.

In order to observe the results more visually, the following evaluation indicators are introduced to compare the results: Mean Absolute Error (MAE), Mean Square Error (MSE), Root-Mean-Square Error (RMSE), and R-square (R^2^).

Due to the small amount of data in this study, only one hidden layer is used to avoid overfitting and reduce the difficulty of model training. The topology of the BPNN in this study is a 11-9-1 structure, as shown in Figure 7. The number of input layer nodes is 11, corresponding to C1, C2, C3, V1, V2, V3, V4, P1, P2, D1, and D2. The number of hidden layer nodes is based on the empirical formula $\sqrt{\mathrm{In}+\mathrm{Out}}+a$ (a ranges from 1 to 10) and is finally determined to be 9 through loop iterations. The number of nodes corresponding to the minimum MSE value in the loop iteration process is taken as the optimal number of hidden layer nodes. The output layer node is 1, corresponding to the diameter of the welding button. In the modeling process, the division ratio of the training set, validation set, and test set is 7:1.5:1.5. The validation set is used to prevent the model from overfitting and improve the generalization ability of the model. The activation functions of the hidden layer and output layer are Tansig and Purelin, respectively. Trainlm is chosen as the iterative algorithm of BPNN.

The prediction of button diameter is carried out using the integrated signal after multi-signal fusion and the single signals before fusion. In order to exclude the influence of the dataset partition on the model simulation effect, ten-fold cross-validation is used. The simulation results are shown in Table 4 in the section Sine-SSA-BP approach. It can be seen from Table 4 that the best simulation effect can be obtained by using the integrated signal combined with the model proposed in this study, and the R^2^ reaches 0.6482. If the Sine-SSA-BP model is used to predict the single signal, the R^2^ of each signal is as follows: the current is 0.5634, the voltage is 0.5532, the power is 0.2373, and the displacement is 0.6324. All of them are smaller than the R^2^ of the integrated signal. This confirms the effectiveness of the signal fusion method proposed in this paper.

### 3.3. Sine-SSA-BP Approach

The sparrow search algorithm is a bionic intelligent optimization algorithm, which is relatively novel and has the advantages of solid optimization ability and fast convergence speed, and it can be summarized as an explorer–follower–warner model [25]. However, as with other swarm intelligence algorithms, the standard SSA will randomly generate individual location information when the program starts running. As a result, the target scheme obtained by the algorithm is not the best and will affect the iterative performance and error rate of SSA [30]. Compared with other maps, the sine map has a simple mathematical form and is convenient to adjust parameters, so this study chooses it as a tool to improve SSA. When α is in the range (3.5, 4], the generated sequences have a better chaotic property [31], and α is set equal to 3.6 in later experiments.

The parameters of the Sine-SSA-BP model are as follows: the population size is 30, the maximum generations are 50, the safety value is 0.6, the proportion of explorers is 0.7, the proportion of warners is 0.2, and the argument limit is 3. The parameters optimized by the Sine SSA algorithm are the weights and thresholds of the hidden layer and output layer of BPNN, the specific realization process of the model is shown in Figure 8, and the algorithm steps are:

Step 1: Data preprocessing. This includes dividing the training and test sets and normalizing the data.

Step 2: Determine the BPNN topology. The nodes of the input layer and output layer are obtained by the size function, and the determination of hidden nodes uses the cycle process, where the minimum error in the cycle process corresponds to the optimal hidden layer node.

Step 3: Initialize BPNN weights and thresholds by sine chaotic map.

Step 4: The Sine-SSA is used to seek the optimal value and threshold.

Step 5: Output BP neural network optimal parameters.

Step 6: Obtain the optimal parameters of the model for instance prediction.

From Table 4, it can be seen that the prediction performance of Sine-SSA-BP is better than that of standard BP for both integrated signal and single signals. This shows that using Sine-SSA can well optimize the standard BP neural network and improve its training effect. The button diameters and errors predicted by the two models of the integrated signal are shown in Figure 9. Figure 9a shows that the diameter predicted by Sine-SSA-BP is closer to the actual value. Compared with standard BP, its overall error fluctuation range is reduced. This indicates that it can better evaluate the button quality.

Figure 9b dissatisfyingly shows that the prediction error fluctuation range of Sine-SSA-BP is still high, and the R^2^ value of ten-fold cross-validation is only 0.6482. The reason is that the dataset is used from an experiment with a fixed set of process parameters, and the active variables of the experiment are the spacing of the spot welds and the gap between the upper and lower sheets. As the disturbance of the environment has little effect on the output current of the welding machine, the change in the experimental conditions only significantly impacts the initial contact resistance. It has a negligible impact on the welding process signals, such as current and voltage. The diameter of the button mainly depends on the magnitude of the welding current. However, the fluctuation of the welding current signal in this study is tiny, which leads to a poor overall prediction effect. However, as shown in Figure 9a, the proposed model can still accurately predict buttons with smaller diameters, which has achieved the purpose of identifying abnormal spot welds, and the accuracy of the test and training set has reached an acceptable range. It is critical for the quality control of resistance spot welding.

### 3.4. Comparative Analysis

Sine-SSA is compared with GA, PSO, SSA, and Sine-PSO on the optimum fitness value of MSE. To compare the algorithm’s performance and optimization speed more objectively, the convergence curve of MSE is drawn according to the evolutionary generations and fitness values, as shown in Figure 10. The optimum MSE value of Sine-SSA is relatively small compared to the other models. This indicates that the model’s ability of finding the global optimal value has been improved. Moreover, its MSE value is already the smallest in the early evolutionary stage, which shows that the convergence speed of the algorithm is faster than the other algorithms.

To evaluate the performance of the RSW quality prediction model objectively, several models are established: GA-BP, PSO-BP, SSA-BP, and Sine-PSO-BP. Based on the integrated signal, the comparative results of different evaluation models are shown in Table 5 and Figure 11. Through comparative analysis, it can be found that the output of other models deviates greatly from buttons with a small diameter (abnormal spot welds). From this perspective, the Sine-SSA-BP model outperforms other models, which is critical for welding quality control.

From the various error indicators in Table 5, it can be seen that the BPNN model can be optimized by various swarm intelligence algorithms. By comprehensive comparison, the Sine-SSA-BP model has the lowest error rate in evaluating the quality of welding spots, and it reduces MAE by 42.33%, MSE by 51.84%, and RMSE by 31.45%. It shows that both SSA and the sine chaotic map can effectively optimize the prediction models based on BP. The R^2^ of the Sine-SSA-BP model reaches 0.6482, while that of BP is only 0.2464, and among all the models listed in Table 5, its R^2^ coefficient is also the highest. The above results show that the proposed model is effective within the range of variables used for training.

## 4. Conclusions

In order to improve the accuracy of welding quality prediction, this paper proposes a method of multi-signal fusion to obtain the integrated signal and uses the Sine-SSA optimized BP neural network to predict welding quality for the first time. The results obtained are as follows:Under the experimental conditions of this study, compared with a single signal, the integrated signal after multi-signal fusion can more accurately predict the button quality of RSW. Combined with the Sine-SSA-BP model, the best prediction effect is obtained. Moreover, abnormal spot welds are successfully predicted, which is critical for welding quality control.The introduction of the sine map can improve the ability of finding the global optimal value of the SSA. The problem of the local optimal solution is avoided. Sine SSA makes it easier and faster to converge to the global optimal MSE value.Sine-SSA is used to optimize the weights and thresholds of a BP neural network, thus predicting the quality of spot welds. The prediction results show that compared with the other models, the Sine-SSA-BP model has better prediction accuracy and is more suitable for the prediction of RSW quality.

## Figures and Tables

**Figure 1 materials-15-07323-f001:**
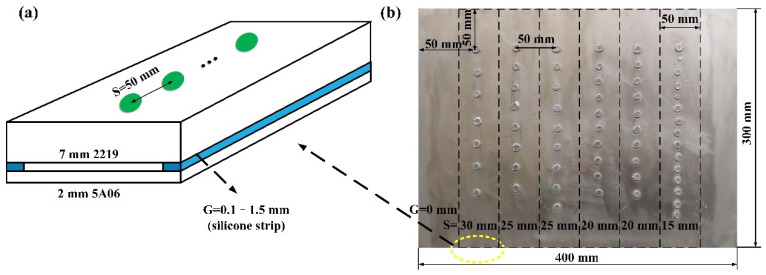
Welding conditions of workpieces and part of stripped weldment: (**a**) different gap conditions, fixed spacing S = 50 mm; (**b**) different spacing conditions, fixed gap G = 0 mm.

**Figure 2 materials-15-07323-f002:**
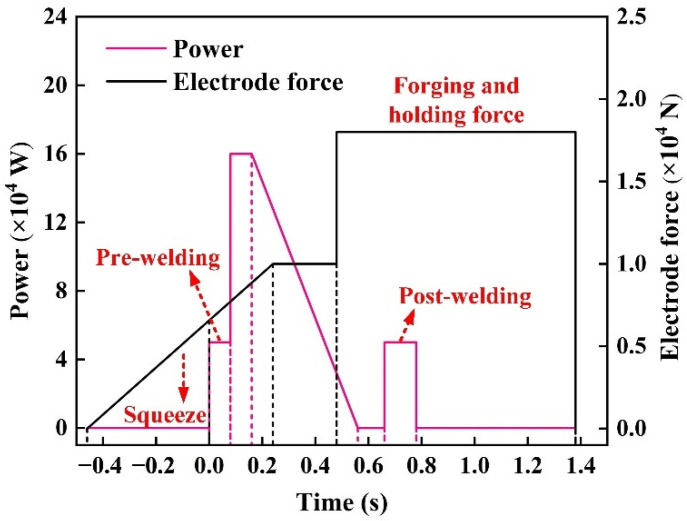
The experimental process parameter curves.

**Figure 3 materials-15-07323-f003:**
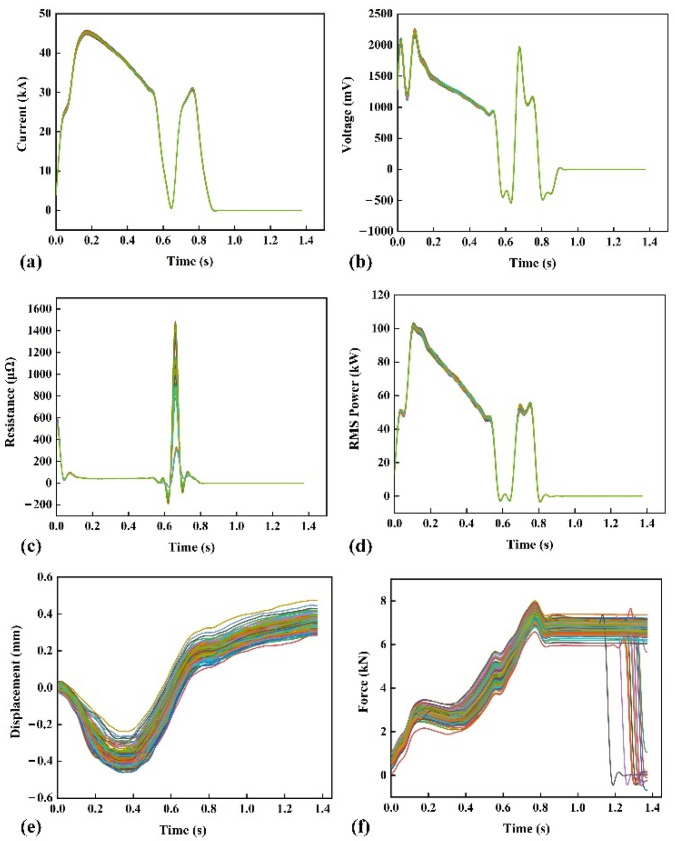
Preprocessed partial welding process signal: (**a**) current; (**b**) voltage; (**c**) dynamic resistance; (**d**) RMS power; (**e**) electrode displacement; (**f**) electrode force.

**Figure 4 materials-15-07323-f004:**
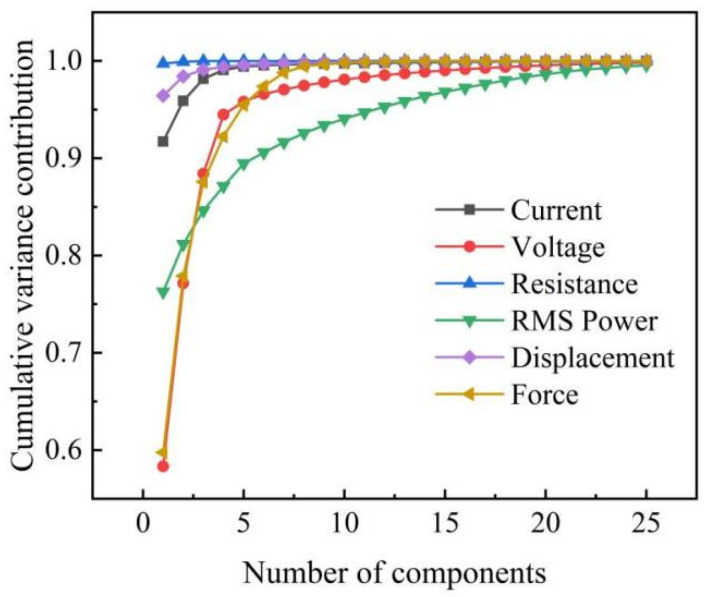
Cumulative variance contribution rate of different signals.

**Figure 5 materials-15-07323-f005:**
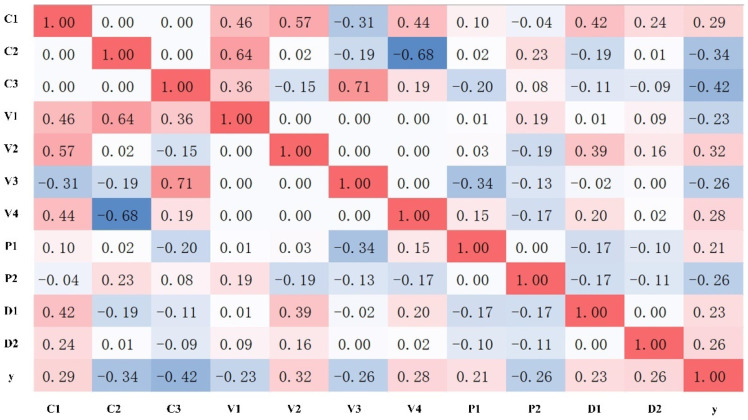
Components with a correlation coefficient greater than 0.2.

**Figure 6 materials-15-07323-f006:**
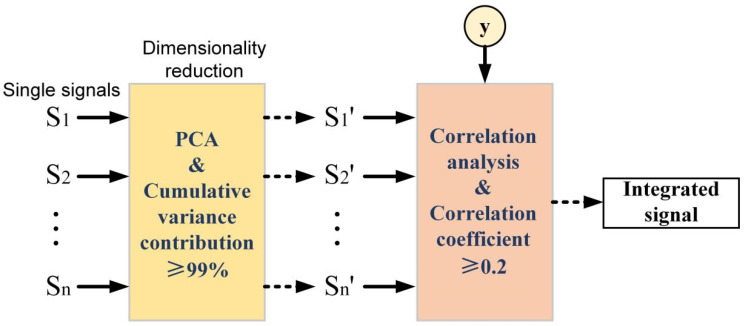
Multi-signal fusion process.

**Figure 7 materials-15-07323-f007:**
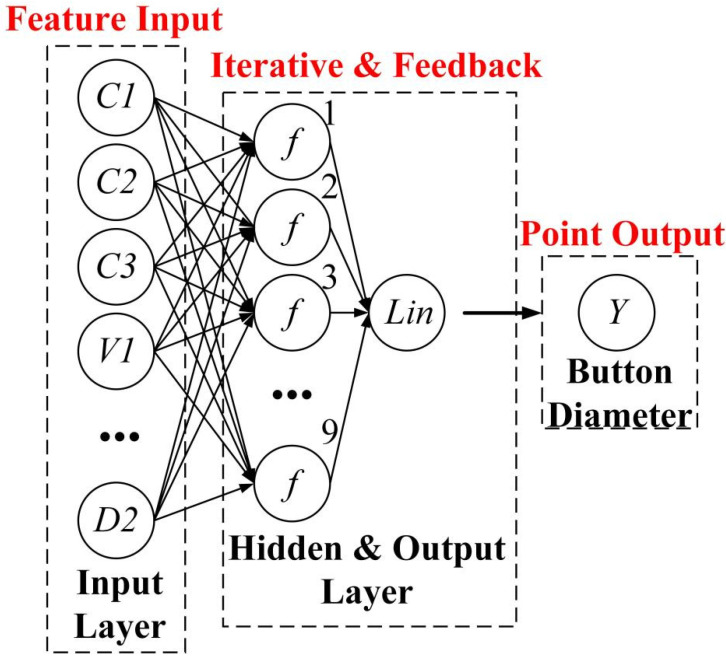
BP neural network 11-9-1 structure diagram.

**Figure 8 materials-15-07323-f008:**
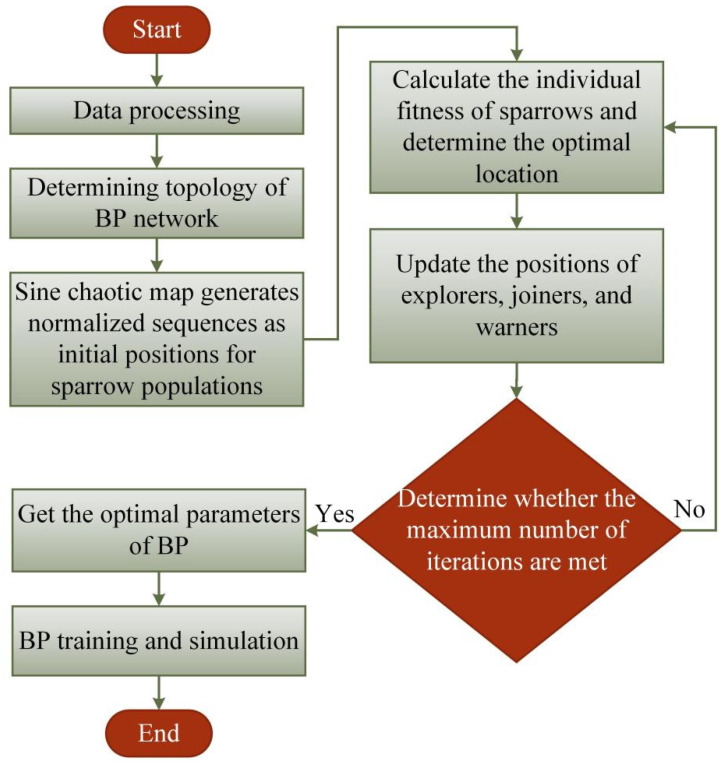
Flowchart of Sine-SSA-BP prediction model.

**Figure 9 materials-15-07323-f009:**
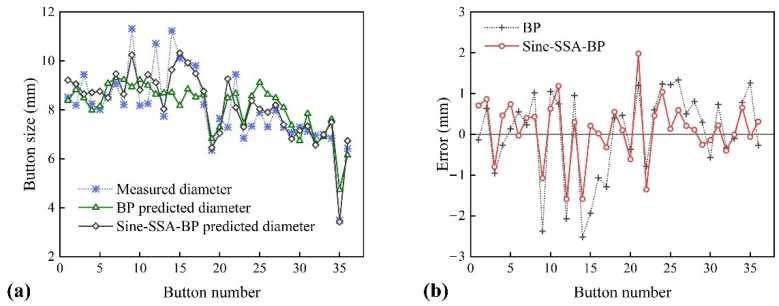
Prediction and error comparison between BP and Sine-SSA-BP model: (**a**) prediction comparison of button size; (**b**) error comparison of BP and Sine-SSA-BP.

**Figure 10 materials-15-07323-f010:**
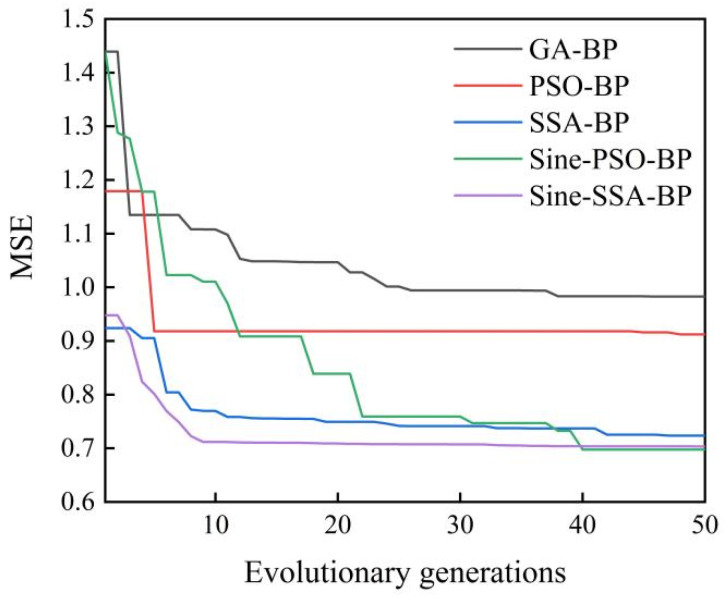
Comparison of evolutionary convergence curves of various algorithms.

**Figure 11 materials-15-07323-f011:**
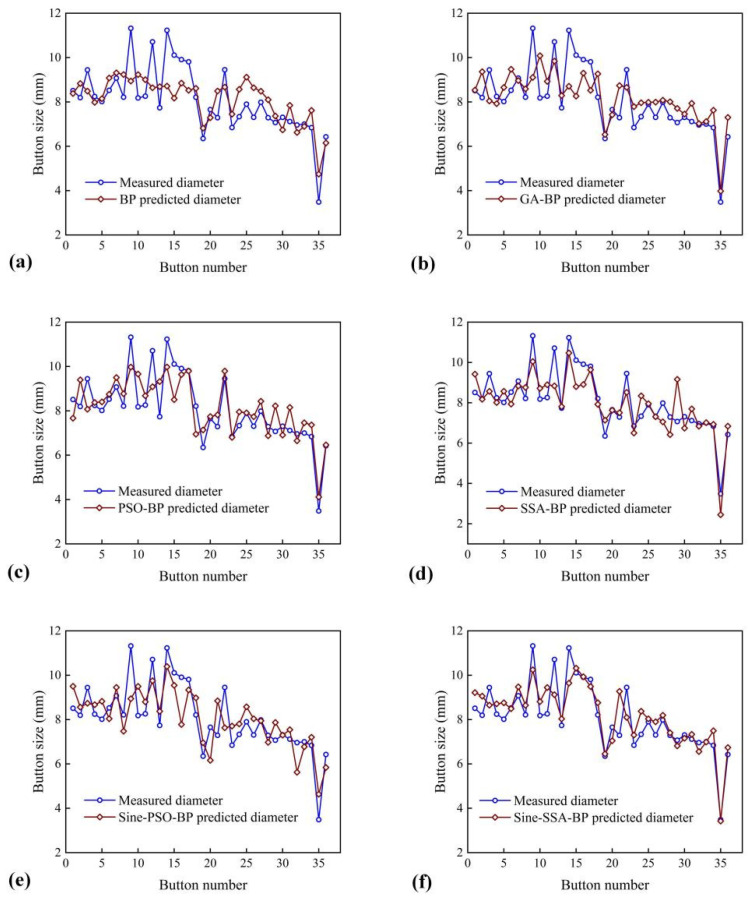
Comparison of measured value and predicted value of various prediction models: (**a**) BP; (**b**) GA-BP; (**c**) PSO-BP; (**d**) SSA-BP; (**e**) Sine-PSO-BP; (**f**) Sine-SSA-BP.

**Table 1 materials-15-07323-t001:** Chemical composition (wt.%) of base materials used in this study.

Materials	Si	Fe	Cu	Mg	V	Mn	Zr	Zn	Ti	Ag	Li	Al
2219	0.06	0.17	6.3	0.02	0.1	0.31	0.15	0.02	0.07	-	-	Bal.
5A06	0.06	0.13	0.03	6.4	-	0.6	-	0.02	0.05	-	-	Bal.

**Table 2 materials-15-07323-t002:** Mechanical properties of base materials used in this study.

Materials	Tensile Strength (MPa)	Yield Strength (MPa)	Elongation (%)
2219	455	366	13.5
5A06	356	187	20.5

**Table 3 materials-15-07323-t003:** Number of spots corresponding to different assembly conditions.

Assembly Condition	Gap (mm)	Spacing (mm)	Number of Replica
1	0.1	50	15
2	0.2	50	15
3	0.3	50	15
4	0.4	50	15
5	0.5	50	15
6	0.6	50	15
7	0.8	50	15
8	1.0	50	15
9	1.5	50	15
10	0	10	15
11	0	15	15
12	0	20	15
13	0	25	15
14	0	30	15
15	0	35	15
16	0	40	15
17	0	45	15
18	0	50	15
Total			270

**Table 4 materials-15-07323-t004:** Comparison of predictions of various signals by Sine-SSA-BP and BP model.

Signals	Predictive Models	Evaluation Indicators			
		MAE	MSE	RMSE	R^2^
Integrated signal	BP	0.9805	1.6618	1.272	0.2464
	Sine-SSA-BP	0.6889	0.7763	0.8719	0.6482
Current	BP	0.9805	1.6455	1.2702	0.2791
	Sine-SSA-BP	0.7404	0.9708	0.9788	0.5634
Voltage	BP	1.0436	1.7738	1.3097	0.2436
	Sine-SSA-BP	0.789	1.0404	1.009	0.5532
RMS Power	BP	1.4859	3.5598	1.8669	−0.5524
	Sine-SSA-BP	1.05	1.7958	1.3246	0.2373
Displacement	BP	0.9375	1.7942	1.3257	0.1957
	Sine-SSA-BP	0.6814	0.788	0.8826	0.6324

**Table 5 materials-15-07323-t005:** Comparison of evaluation indicators among prediction models.

Prediction Models	Evaluation Indicators			
	MAE	MSE	RMSE	R^2^
BP	0.9805	1.6618	1.272	0.2464
GA-BP	0.8336	1.172	1.077	0.4762
PSO-BP	0.8011	1.0517	1.0194	0.5402
SSA-BP	0.6869	0.8099	0.8908	0.6458
Sine-PSO-BP	0.7754	1.0237	0.991	0.5728
Sine-SSA-BP	0.6889	0.7763	0.8719	0.6482

## Data Availability

Data sharing is not applicable to this article.

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
