# Peer review of "Prediction of Resistance Spot Welding Quality Based on BPNN Optimized by Improved Sparrow Search Algorithm"

_materials, 2022, doi:10.3390/ma15207323_

Round 1
Reviewer 1 Report
The presented manuscript seems to be interesting for readers of the Materials journal, it is written in a good manner and suits the requirements of the journal. It can be accepted for publication after minor corrections listed below.
- English language of manuscript is acceptable in general. However, it would be much better to improve. Please avoid the unnecessary long sentence. Also, some grammatical and typos mistakes can be observed. For example: eexperimental , interval (3.5, 4]
- Please respond to the following questions and make necessary revisions.
- Is the data properly divided as testing (20-30%) and training (70-80%) sets?
- Are the accuracies of testing and training sets within acceptable ranges?
- Is it stated that the proposed models will be valid within the ranges of variables used for training?
- Figure 8 is repeated twice. “Figure 8. Prediction and error comparison…” should be changed to “Figure 9” and the rest of the figures should be modified in the same way.
-The composition, structure and properties of sheets (2219 and 5A06) and their dimensions and thickness should be given in text
- Figure 2(a) has no special scientific value and can be omitted
- Acronyms, such as PCA and RMS, should all be defined at their first occurrence in the manuscript; It is suggested to add a section for the acronyms and parameters at the end of the manuscript.
- Literature review is not sufficient and authors must review and cite more papers in the field of correlation and prediction of the structure and properties of steels and especially newly published ones. Doing this, review and citing the following refs could be helpful:
[] Journal of Mining and Metallurgy, Section B: Metallurgy, 51, 2015, 173-178.
[] Measurement, 99, 2017, 120-127.
Reviewer 2 Report
Major revision is required.
Please, revise the paper based on the attached pdf.

Round 2
Reviewer 1 Report
As authors have performed an adequate revise, the manuscript might be accepted for publication in the journal of Materials.
Reviewer 2 Report
Accept in present form.